# Comparison of Quality, Antioxidant Capacity, and Anti-Inflammatory Activity of Adlay [*Coix lacryma-jobi* L. var. *ma-yuen* (Rom. Caill.) Stapf.] Sprout at Several Harvest Time

**DOI:** 10.3390/plants12162975

**Published:** 2023-08-17

**Authors:** Eun-Song Lee, Yong-Il Kim, Jeong-Hoon Lee, Yong-Goo Kim, Kyung-Sook Han, Young-Ho Yoon, Byoung-Ok Cho, Kyungtae Park, Hamin Lee, Ju-Sung Cho

**Affiliations:** 1Department of Herbal Crop Research, National Institute of Horticultural & Herbal Science, Eumseong 27709, Republic of Korea; eslee24@korea.kr (E.-S.L.);; 2Division of Animal, Horticultural and Food Sciences, Chungbuk National University, Cheongju 28644, Republic of Korea; 3Institute of Health Science, Jeonju University, Jeonju 55069, Republic of Korea; 4Brain Korea 21 Center for Bio-Health Industry, Chungbuk National University, Cheongju 28644, Republic of Korea

**Keywords:** adlay, harvest, plant-based foods, physiological effects, sprout vegetables

## Abstract

Recently, there has been a growing interest in the consumption of plant-based foods such as vegetables and grains for the purpose of disease prevention and treatment. Adlay seeds contain physiologically active substances, including coixol, coixenolide, and lactams. In this study, adlay sprouts were cultivated and harvested at various time points, specifically at 3, 5, 7, 9, and 11 days after sowing. The antioxidant activity of the extracts was evaluated using assays such as DPPH radical scavenging, ABTS radical scavenging, reducing power, and total polyphenol contents. The toxicity of the extracts was assessed using cell culture and the WST-1 assay. The aboveground components of the sprouts demonstrated a significant increase in length, ranging from 2.75 cm to 21.87 cm, weight, ranging from 0.05 g to 0.32 g, and biomass, ranging from 161.4 g to 1319.1 g, as the number of days after sowing advanced, reaching its peak coixol content of 39.38 mg/g on the third day after sowing. Notably, the antioxidant enzyme activity was highest between the third and fifth days after sowing. Regarding anti-inflammatory activity, the inhibition of cyclooxygenase 2 (COX-2) expression was most prominent in samples harvested from the ninth to eleventh days after sowing, corresponding to the later stage of growth. While the overall production mass increased with the number of days after sowing, considering factors such as yield increase index per unit area, turnover rate, and antioxidant activity, harvesting at the early growth stage, specifically between the fifth and seventh days after sowing, was found to be economically advantageous. Thus, the quality, antioxidant capacity, and anti-inflammatory activity of adlay sprouts varied depending on the harvest time, highlighting the importance of determining the appropriate harvest time based on the production objectives. This study demonstrates the changes in the growth and quality of adlay sprouts in relation to the harvest time, emphasizing the potential for developing a market for adlay sprouts as a new food product.

## 1. Introduction

Recently, interest in plant food intake for disease prevention and treatment has increased. Among plant food, sprout vegetables have garnered attention as a health food [1]. The development of functional food materials using sprouts, such as radish [2], barley [3], peanut [4], and ginseng [5], is being actively investigated. Sprouts generally refer to the state of young cotyledons approximately a week after germination, at which time they contain a large amount of dietary fiber, including various amino acids, enzymes, vitamins, and minerals [6]. In addition, fat and calories are lowered during the germination process, and various physiologically active substances are produced to protect the seeds from external attacks [7].

The production period of sprouts is very short (about a week). Therefore, it is important to identify the optimal production conditions. Suitable seed sizes, segregation of ripe seeds, and improved germination rates for commercial sprout hemp production were recently studied [8]. Another study was conducted on the climatic factors crucial to produce barley sprouts rich in bioactive saponarin [9]. Recently, as the sprouting market has been spotlighted, research on production conditions for peanuts, alfalfa, soybeans, radishes, and broccoli has been conducted [10,11,12].

Extensive research has been undertaken to cultivate sprouts with enhanced functionality, and in broccoli and radish sprouts, the glucosinolates’ content was increased through elicitation and seed priming [13]. In lentil sprouts, a study on the increase in nutrients through elicitation with abiotic stresses has been conducted [14], while in pea sprouts, research was conducted to increase phenolic compounds and antioxidant activities through LED [15]. Other studies on increasing nutritional components in sprouts and microgreens through LED have been conducted [16].

Coix (*Coix lachryma-jobi* L.), also referred to as adlay or Job’s tears, is a lesser-known cereal that holds significant culinary value in certain Asian regions [17]. Coix seeds, originating from tropical Asia, contain a significant amount of starch in their endosperm along with polysaccharides, proteins, and seed oil, among other chemical components [18].

The anticancer effect of adlay stems and leaves extract [19], antidiabetic activity of adlay seeds extract [20], improvement of inflammation and oxidative stress in a rheumatoid arthritis model [21], preventive effect on nonalcoholic fatty liver disease [22], cell cycle arrest effect in breast and cervical cancer cell lines [23,24], high-fat diet-induced mice, attenuation of obesity [25], and effect of Cohlrax’s diet on obesity and hyperlipidemia in mice fed a high-fat diet were investigated [26]. And the contents of riboflavin and coixol in adlay sprout were 2,2-di(4-tert-octylphenyl)-1-picrylhydrazyl (DPPH) (−0.743, −0.969) and 2,2′-azinobis-(3-ethylbenzothiazoline-6-sulfonic acid (ABTS) (−0.796, −0.958), which were significantly correlated with the IC_50_ value of clearance [27].

The active ingredient of adlay is coixol, which suppresses nuclear transcription factor κ B (NF-κB), mitogen-activated protein kinases (MAPK) pathways, and NOD-like receptor protein 3 (NLRP3) inflammasome activation in lipopolysaccharide-induced RAW 264.7 cells [28]. Adlay sprouts were investigated for their in vitro anti-colon cancer activity [29]. Through HPLC–DAD–ESI-MS/MS analysis, it was determined that ferulic acid was the most abundant phenolic acid in the fraction of defatted adlay seed meal, with a concentration of 67.28 mg/g. And rutin, on the other hand, was identified as the predominant flavonoid, with a content of 41.11 mg/g [30]. Furthermore, it has been suggested that coix seed may impact human immune function [31]; in addition, the polyphenol extract of adlay exhibited notable hypocholesterolemic and antioxidant activities, potentially contributing to its protective effects on cardiovascular health in vivo [12].

Adlay seeds with excellent functionality have been reported to efficiently improve nutritional compounds while germinating and altering their physicochemical properties [32]. Germinated adlay water extract showed higher inhibitory activity against angiotensin-converting enzyme (ACE), xanthine oxidase, and tyrosinase than non-germinated adlay water extract, showing the potential to improve the nutritional value and physiological activity of adlay sprout extract [33]. The objective of this study was to compare the quality, antioxidant capacity, and anti-inflammatory activity of adlay sprouts, utilizing the aboveground portion as food material, at various harvest times.

## 2. Results

### 2.1. Growth Characteristics, Functionality, and Nutritional Content of Adlay Sprout

Adlay sprouts exhibited the initial signs of germination as light green sprouts on the second day after sowing. By the third day, the sprouts had developed into a shape with an aboveground part measuring approximately 2.75 cm in length (Figure 1). Investigation into the growth characteristics of adlay sprouts revealed a significant increase in both the length and weight of the aboveground parts as the harvesting time progressed from 3 to 11 days after sowing (Table 1). Specifically, the aboveground part length expanded from 2.75 cm to 21.87 cm over the course of 3 to 11 days after sowing, while the weight of the sprouts increased from 0.05 g to 0.32 g. Additionally, the number of sprouts rose from 161.4 g to 1319.1 g. 

The initial coixol content of adlay seeds before sowing (0 day) was measured at 0.11 mg/g. However, within just three days after sowing, the coixol content exhibited a remarkable increase of more than 350 times, reaching 39.38 mg/g. A comparison between the highest content observed on the third day (39.38 ± 0.04 mg/g) and the lowest content on the eleventh day (19.08 ± 0.01 mg/g) revealed that the former was approximately twice as high as the latter.

As the harvesting period of adlay sprouts increased, the brightness (L) showed a significant tendency to become brighter. Additionally, as the growth period extended, the redness tended to decrease, while no significant difference was observed in yellowness.

Regarding the content of free sugars and sugar alcohols in adlay sprouts at different harvest times (as shown in Table 2 and Table 3), it was observed that the glucose content was relatively high at 74.66 mg/g on the third day of growth. As the growth period advanced, the levels of glucose, galactose, and fructose initially increased and then decreased. On the other hand, mannitol and inositol levels increased towards the latter part of the growth period.

### 2.2. Antioxidant Activity

To assess the toxicity of adlay extracts at each growth stage, a WST-1 assay was performed on the human hepatocellular carcinoma cell line (HepG2) and RAW264.7 cell lines. The results indicated that there was no observed toxicity effect up to a concentration of 100 µg/mL of the extract (Appendix A).

The antioxidant activity of adlay sprouts at different harvest times was evaluated through various assays, including DPPH and ABTS radical scavenging activities, reducing power, and total polyphenol content (Figure 2). The adlay sprouts harvested on the third day after sowing exhibited the highest DPPH radical scavenging activity (370.40 µg/mL) and ABTS radical scavenging activity (594.85 µg/mL). Additionally, they demonstrated a high polyphenol content of 23.42 mg GAE/g. The reducing power assay indicated that the sprouts harvested on the ninth day after sowing exhibited the highest reducing power.

In the HepG2 cell line, the inhibitory activity of adlay sprout extract against reactive oxygen species (ROS) generation, induced by intracellular oxidative stress, was investigated using a flow cytometer (Figure 3a). The seed samples, as well as the sprout samples harvested on the seventh and eleven days after sowing, and the comparative drug NAC, showed a tendency to reduce intracellular ROS.

To determine the activity of antioxidant enzymes in response to oxidative stress induced by hydrogen peroxide in HepG2 cells, the activities of superoxide dismutase (SOD) and catalase (CAT) were measured (Figure 3b,c). The sample extracts from the third, fifth, and ninth days after sowing exhibited high SOD activity, while catalase activity tended to increase in the sample extracts from the third, fifth, ninth, and eleven days after sowing.

### 2.3. Anti-Inflammatory Activity

In RAW264.7 cells treated with the endotoxin LPS, the adlay extract’s ability to inhibit the increased production of nitric oxide (NO) during inflammatory reactions was evaluated using the Griess assay. The results indicated that all samples, except the seed (day 0) sample, exhibited more than 50% inhibition of NO (Figure 4a).

Furthermore, an ELISA assay was conducted to assess the inflammatory cytokine inhibitory ability of adlay sprout extracts harvested at different times in RAW264.7 cells treated with LPS. The extracts from the fifth, seventh, ninth, and eleven days showed more than 50% inhibition of IL-6 expression (Figure 4b). Notably, all extracts, except the 0 day (adlay seed) sample, exhibited an inhibitory effect of more than 50% on PGE2 (Figure 4c).

Western blotting was conducted to assess the phosphorylation inhibitory ability of NF-κB, an anti-inflammatory signaling factor, in LPS-treated RAW264.7 cells at different harvest times. The samples obtained on the ninth and eleventh days after sowing exhibited a high inhibitory ability (Figure 5).

Furthermore, Western blotting was performed to measure the anti-inflammatory effect of adlay sprout extract harvested at different times in LPS-treated RAW264.7 cells. The extracts showed an increase in the expression of the antioxidant enzyme HO-1 with increasing harvest time, with the order of expression being 7 < 9 < 11 days after sowing (Figure 6a,b). Additionally, the extracts obtained from the seeds (day 0), ninth, and eleventh days showed inhibition of COX-2 expression (Figure 6a,c), while extracts from the third and seventh days exhibited high expression inhibition of iNOS (Figure 6a,d).

## 3. Discussion

During seed germination, young shoots undergo a process where they reduce fat and calories while producing a range of bioactive substances to defend against external threats [7]. Studies on peanuts have shown that sprouts have higher resveratrol content compared to seeds [34]. Additionally, barley sprouts have been found to contain significant levels of polycosanol and saponarin, which are effective in preventing alcoholic fatty liver disease [35]. These findings highlight the nutritional and medicinal value of sprouts and their potential for promoting health and preventing diseases.

Coixol, found in coix seeds, has a long history of traditional use for treating various conditions such as cancer, warts, and skin pigmentation [31]. It has been observed that coixol levels increase by 3.6 times in germinated seeds compared to conventional seeds after 60 h of germination [32]. In this study, coixol components were not only detected in the seeds but also in the sprouts, with the sprouts containing higher levels of coixol components than the seeds.

In the case of barley sprouts, chromaticity, which includes measures of lightness, redness, and yellowness, has been used as an indicator [36]. This measurement index allows for the assessment of color characteristics in barley sprouts, providing valuable information on their quality and visual appeal.

Recently, there has been a growing interest in plant-derived natural sugars due to their potential health benefits. Stevia (*Stevia rebaudiana* Bertoni) a perennial herb from the Asteraceae family, contains stevioside in its stems and leaves, which is a natural sweetener that is 200 to 300 times sweeter than sugar [37,38]. Similarly, studies have been conducted on the functional properties of fructo-oligosaccharides, a natural sugar component found in yacon, known for its potential health benefits [39,40,41].

In the case of ginseng sprouts, changes in ginsenoside content were observed based on harvest time. The total ginsenoside content in the stem decreased after transplanting the seedlings, while the content in the root increased up to 21 days and then rapidly declined [5].

Considering the growth characteristics, functionality, and nutritional components of adlay sprouts, the optimal harvesting time was found to be between 3 and 5 days after sowing. The yield increase index (the current production mass divided by the previous production mass) showed a gradual decrease as the number of days after sowing increased, indicating that early harvesting is more efficient in terms of production. Harvesting on the ninth day after sowing resulted in high total sugar alcohol content but a low yield increase index, indicating inefficiency in production. Therefore, harvesting adlay sprouts on days 5 to 7 was deemed appropriate considering the yield increase index.

Intracellular reactive oxygen species (ROS) are naturally generated during metabolic processes but can become harmful when their balance is disrupted. Excessive ROS can lead to DNA damage, protein denaturation, cell membrane oxidation, and inflammation, ultimately causing cell death. In this context, it is crucial to inhibit the formation of intracellular ROS to prevent apoptosis [42]. The antioxidant activity analysis revealed that adlay sprouts harvested on the third day after sowing exhibited the highest antioxidant activity.

Previous studies have shown that germinated adlay seeds exhibit increased levels of free phenols and flavonoids, while bound phenols and flavonoids are significantly reduced after germination [43]. Furthermore, germinated adlay seeds treated with citric acid have shown a significant increase in phenol (18.3%) and flavonoid (17.0%) content, as well as antioxidant activity (39.1%) compared to the control [44].

Hydrogen peroxide (H_2_O_2_) is a major compound produced during the oxidative burst in plants and can easily pass through plant cell membranes, acting as a relatively lasting signal [45]. There are many ways to produce H_2_O_2_ in plant cells, a significant part of which is produced by the imbalance of O^˙2−^ by SOD, and the accumulation of O^˙2−^ is the main cause of cellular lipid peroxidation [46]. CAT is the primary enzyme responsible for the removal of H_2_O_2_.

In the case of other sprout crops, studies have demonstrated the protective effects against oxidative stress in hepatocytes by activating Nrf2 and increasing glutathione synthesis in sprouted barley [47]. In buckwheat sprouts, artificial light has been used to increase antioxidant activity and flavonoid content [48]. Similarly, the antioxidant activity of adlay sprout extract was confirmed in HepG2 cells treated with hydrogen peroxide, indicating its ability to protect against oxidative stress.

Barley sprouts, like adlay sprouts, have shown significant inhibition of iNOS and COX-2 expression in RAW 264.7 cells previously stimulated with LPS, indicating their anti-inflammatory activity [49]. Adlay seed extract (ASE) has been reported to be effective in improving rheumatoid arthritis in mice [21]. Coixol, a plant polyphenol extracted from adlay, has been found to inhibit the activation of MAPK (mitogen-activated protein kinases), NF-κB pathway, and NLRP3 inflammasome, exerting specific anti-inflammatory effects by suppressing the expression of pro-inflammatory mediators in vitro [28]. Additionally, a previous study demonstrated that hot water extract of adlay sprout improved ulcerative colitis induced by DSS in an animal model.

These findings collectively suggest that adlay sprouts possess antioxidant and anti-inflammatory properties, making them a potential functional food ingredient with beneficial effects on oxidative stress-related conditions and inflammatory diseases.

## 4. Materials and Methods

### 4.1. Adlay Sprout and Extract

The adlay seeds used in this experiment were obtained from the harvest in September 2021, in Yeoncheon-gun, Gyeonggi-do, Korea. The seeds were stored at a low temperature of 4 °C until use. To achieve the desired initial moisture content, the primed seed material was immersed in distilled water at room temperature for 24 h and subsequently dried for 48 h at 20 °C using an oven. Sprouts were cultivated in a growth chamber under controlled light conditions (metal halide lamp condition) at a temperature of 27 °C and relative humidity of 98% according to the method of previous experiment (Appendix A). The presence or absence of light was considered during the evaluation of sprout production mass, as outlined in Appendix A. Following established procedures from previous studies, the seeds were sown at an appropriate density of 800 g/cell in a planting area of 60 × 30 cm (Appendix A). Watering was carried out three times per day, as depicted in Appendix A. The initial seed prior to germination is regarded as the primary source for the 0 day sample. The aboveground parts of the sprouts were randomly harvested at 3, 5, 7, 9, and 11 days after sowing to measure sprout length, weight, and production mass. To prepare the adlay sprout extract for experimental purposes, it was diluted in 70% EtOH at a ratio of 1:40 (*w/v*). Subsequently, the extract was concentrated under reduced pressure and subjected to freeze-drying using a freeze dryer (Operon Co., Ltd., Gimpo-si, Gyeonggi-do, Republic of Korea).

### 4.2. Measurement of Functional Components and Color

#### 4.2.1. Coixol Content Analysis

The analysis of coixol content of adlay sprout extract was performed by partially modifying the analytical method of previous studies [27,50]. The adlay sprouts were extracted by adding 70% ethanol at a ratio of 40 times the weight of the sample. The extraction was carried out for three days at 50 degrees Celsius using a Floor standing shaking incubator (JSSI-300C, JSR, Gongju-si, Chungcheongnam-do, Republic of Korea). After extraction, the resulting extract was filtered three times using a filter paper and a paper filter. The filtered extract was then concentrated using a rotary evaporator (R-100; BUCHI, Flawil, Switzerland) under reduced pressure and subsequently freeze-dried for use in the experiment.

The lyophilized powder of adlay sprout extract was dissolved in methanol to achieve a concentration of 10 mg/mL. The resulting solution was then passed through a 0.45 μm PTEF filter (Gelman, New York, NY, USA). Subsequently, HPLC analysis was performed using a YMC-Triart C18 column (250 mm × 4.6 mml.D., S-5 um, 12 nm) (YMC, Kyoto, Japan) maintained at a temperature of 40 °C (Table 1). The mobile phase was composed of 0.1% formic acid in water (A) and 0.1% formic acid in acetonitrile (B). The concentration gradient was adjusted over time according to the schedule provided in Table 4. The flow rate was set at 0.8 mL/min, and the absorbance of the compounds was detected at 230 nm. Ten μL of the sample solution was injected into the HPLC system (e2695; Waters Co., Milford, MA, USA) for analysis. To determine the content of coixol in the extract, a standard calibration curve was generated by gradually diluting the coixol standard from 1 to 200 μg/mL. The extract content was quantified by calculating the area-to-concentration ratio based on this calibration curve.

#### 4.2.2. Chromaticity Measurement

The chromaticity analysis was performed using a color difference meter (CM-2600d; Konica Minolta, Tokyo, Japan). Three measurements were taken for the L value (lightness), a value (redness), and b value (yellowness), and the average values were calculated. The standard whiteboard used for reference had the following chromaticity values: L-value of 99.13, a-value of 0.01, and b-value of 0.14.

#### 4.2.3. Analysis of Free Sugar and Sugar Alcohol Content

To analyze the components of free sugar and sugar alcohol content of the adlay sprout extract, BSTFA (N,O-bis(trimethylsilyl)trifluoroacetamide) was used for derivatization, converting the polar functional groups such as amine, -OH, and -COOH into -OSi(CH_3_)_3_ for GC/MS analysis. This derivatization process enhanced the ease of analysis. The structures of the analyzed free sugars and sugar alcohols are shown in Figure 7. The GC/MS analysis was performed using a GC/MS QP 2010 plus instrument (Shimadzu, Kyoto, Japan) equipped with an Rxi-5MS column (0.25 mm × 30 m, film thickness 0.25 μm). The injector temperature was set at 300 °C. The oven temperature was initially held at 70 °C for 3 min, then increased at a rate of 10 °C/min until reaching 320 °C, where it was maintained for 5 min. Helium (He) was used as the carrier gas at a flow rate of 1 mL/min, and 1 μL of the sample was injected with a split ratio of 10:1. The GC/MS interface temperature was set to 230 °C, and electron ionization (EI) was performed at an ionization voltage of 70 eV. The SCAN mode was used to analyze the 45–550 m/z range for molecular weight, and the materials were qualitatively analyzed by evaluating the fragmentation patterns of each component.

The results of qualitative analysis were compared with the composition of the database library within the analysis equipment. In order to perform quantitative analysis of free sugars and sugar alcohols, the resolution of each standard substance was checked, and the linearity of the calibration curves for each component was confirmed. The standard substances used for quantitative analysis were fructose, galactose, glucose, maltose, and lactose for free sugars, and glycerol, xylitol, arabitol, mannitol, sorbitol, and inositol for sugar alcohols. The determination coefficients (r^2^) of the calibration curves for each component were all found to be above 0.99.

### 4.3. Determination of Antioxidant Activity

#### 4.3.1. Experimental Material

Dulbecco’s Modified Eagle Medium (DMEM), Fetal Bovine Serum (FBS), and penicillin-streptomycin were purchased from Gibco (Thermo Fisher Scientific, Waltham, MA, USA). Potassium persulfate, 2,2′-Azino-bis(3-ethylbenzothiazoline-6-sulfonic acid) (ABTS), 2,2-Diphenyl-1-picrylhydrazyl (DPPH), and Folin & Ciocalteu’s phenol reagents were purchased from Sigma-Aldrich (St. Louis, MO, USA). Carboxy-H_2_DCFDA was purchased from Invitrogen (Thermo Fisher Scientific, Waltham, MA, USA), while superoxide dismutase (SOD) and catalase assay kits were obtained from Cayman (Cayman Chemical Company, Ann Arbor, MI, USA).

#### 4.3.2. 2,2-Diphenyl 1 Picrylhydrazyl (DPPH) Radical Scavenging Activity

The DPPH radical scavenging activity was measured using the method described in previous study [51]. The sample was dissolved in ethanol and quantified to a final concentration of 0 to 1000 µg/mL, 100 µL of each sample was injected into a 96 well plate, and 100 µL of 0.3 mM DPPH was added at the same time to obtain a total volume of 200 µL. After incubating at room temperature for 10 min, the absorbance was measured at a wavelength of 540 nm using an ELISA reader (Tecan Group Ltd., Mannedorf, Switzerland). The DPPH radical scavenging ability was expressed as a percentage of the difference in absorbance between the sample solution added group and the non-added group using the following formula:DPPH radical scavenging ability (%) = {1 − (absorbance of added group/absorbance of non-added group)} × 100

The electron-donating capacity was expressed as the IC_50_ value, representing the concentration (µg/mL) of the sample required to reduce the DPPH radical by 50%

#### 4.3.3. ABTS (2,2′-azino-bis (3-ethylbenzothiazoline-6-sulfonic Acid))

ABTS (7.4 mM) and potassium persulfate (2.6 mM) were mixed and incubated in the dark at 4 °C for 24 h. Subsequently, the mixture was diluted with distilled water to achieve an absorbance of ±0.04 (±standard error) at 732 nm. After quantifying each sample, 50 µL of sample was dispensed into a microtube, and 950 µL of the ABTS solution was dispensed into the microtube, mixed and incubated for 30 min in the dark. Hundred µL of the reaction was then dispensed into a 96 well plate, and absorbance was measured at 732 nm measured with an ELISA reader. ABTS scavenging activity was calculated as a percentage of the difference in absorbance between the groups with and without the addition of the sample solution.
ABTS radical scavenging activity (%) = {1 − (absorbance of added group/absorbance of non-added group)} × 100

The electron-donating capacity was expressed as the IC_50_ value, representing the concentration (µg/mL) of the sample required to reduce the ABTS radical by 50%

#### 4.3.4. Reducing Power

The lyophilized powder of adlay sprout extract was dissolved in distilled water and adjusted to a final concentration ranging from 0 to 1000 µg/mL. Each sample (250 µL) was mixed with 0.2 M sodium phosphate buffer and 1% potassium ferricyanide in a microtube. The mixture was then incubated at 50 °C for 20 min and subsequently cooled to room temperature. Next, 250 µL of 10% trichloroacetic acid was added to the mixture, followed by centrifugation at 1420× *g* for 10 min. After centrifugation, 500 µL of the supernatant, 500 µL of DW, and 100 µL of 0.1% FeCl_3_ were combined. Subsequently, 200 µL of each sample was dispensed into a microplate, and the absorbance was measured at a wavelength of 700 nm using an ELISA reader (Tecan Group Ltd., Mannedorf, Switzerland). As a control, butylated hydroxytoluene (BHT) was used, and the reducing power was quantified based on the absorbance value.

#### 4.3.5. Total Polyphenol Contents Assay

The freeze-dried powder of adlay sprout extract was quantified to a final concentration of 250 µg/mL by reconstitution in distilled water. A total of 100 µL of each sample and Folin-reagent were added to microtubes and allowed to react at room temperature for 5 min. Then, 1 mL of 4% Na_2_CO_3_ was added, and after 90 min, 200 µL aliquots were dispensed into a microplate for measurement of absorbance at 700 nm wavelength using an ELISA reader (Tecan Group Ltd., Mannedorf, Switzerland). The measured values were used to calculate the content (mg GAE/g) based on a standard curve.

#### 4.3.6. ROS Evaluation

The HepG2 cell line was obtained from the Korea Cell Line Bank. HepG2 cells were dispensed into a 6-well plate at a final concentration of 4 × 10^5^ cells/mL. Following a 24-h incubation at 37 °C and 5% CO_2_, the samples were treated with a concentration of 100 µg/mL. After 1 h, tert-butyl hydroperoxide (tBHP) was treated to a concentration of 200 µM and cultured for 6 h. Thereafter, Carboxy-H_2_DCFDA (Thermo Fisher Scientific, Waltham, MA, USA) was injected into the plate at a concentration of 5 μM and incubated at 37 °C for 30 min. The cells were then detached with trypsin and suspended in phosphate-buffered saline (PBS), and intracellular ROS levels were measured using flow cytometry (Cyto FLEX LX; Beckman Coulter, Inc., Indianapolis, IN, USA).

#### 4.3.7. Superoxide Dismutase (SOD) Activity Assay

After dispensing HepG2 cells to a final concentration of 4 × 10^5^ cells/mL in a 6-well plate, the plate was incubated for 24 hat 37 °C and 5% CO_2_. The sample was treated at a concentration of 50 µg/mL, and after 1 h, H_2_O_2_ (1 mM) was added and incubated for 3 h. Thereafter, the proteins were collected, quantified, and tested using an SOD activity assay kit (Cayman Chemical Company, Ann Arbor, MI, USA).

#### 4.3.8. Catalase Activity Assay

After dispensing HepG2 cells to a final concentration of 4 × 10^5^ cells/mL in a 6-well plate, the plate was incubated for 24 hat 37 °C and 5% CO_2_. The sample was treated at a concentration of 50 µg/mL, and after 1 h, H_2_O_2_ (1 mM) was added and the plate was further incubated for 3 h. Thereafter, the proteins were collected, quantified, and tested using a catalase assay kit (Cayman Chemical Company, Ann Arbor, MI, USA).

### 4.4. Toxicity Evaluation

#### 4.4.1. Cell Culture

HepG2 cells were cultured in DMEM (GIBCO-BRL, Invitrogen, Carlsbad, CA, USA), containing 10% FBS and 1% antibiotics (penicillin and streptomycin) in a 5% CO_2_ incubator (MCO-15AC, SANYO, Osaka, Japan) maintained at 37 °C with sufficient humidity.

#### 4.4.2. WST-1 Assay

After dispensing HepG2 cells to a final concentration of 2 × 10^5^ cells/mL in a 96-well plate, incubating them in an incubator at 37 °C and 5% CO_2_ for 24 h, the samples were treated at concentrations of 25, 50, and 100 µg/mL, and incubated for 20 h. After injecting 10 µL of Quanti-MaXTM into the plate, the plate was incubated for 4 h. Absorbance was measured at a wavelength of 450 nm using an ELISA reader.

### 4.5. Anti-Inflammatory Evaluation

#### 4.5.1. Experimental Materials

For the anti-inflammatory evaluation of the sprouted adlay samples, Dulbecco’s Modified Eagle Medium (DMEM), Fetal Bovine Serum (FBS), and penicillin–streptomycin were purchased from Gibco, a brand of Thermo Fisher Scientific located in Waltham, MA, USA. Lipopolysaccharides were obtained from Sigma-Aldrich in St. Louis, MO, USA. The IL-6 and PGE2 ELISA kit were purchased from R&D Systems based in Minneapolis, MN, USA. Antibodies against HO-1 and iNOS were obtained from BD Biosciences located in Franklin Lakes, NJ, USA. The anti-cyclooxygenase 2 (COX-2) antibody was obtained from Cayman Chemical Company in Ann Arbor, MI, USA. Antibodies against p-NF-κB, NF-κB, and β-actin were purchased from Santa Cruz Biotechnology (Santa Cruz, CA, USA).

#### 4.5.2. Cell Culture

The RAW264.7 cells, a mouse-derived macrophage cell line used in this experiment, were obtained from ATCC (American Type Culture Collection, Manassas, VA, USA). The cells were cultured in DMEM (GIBCO-BRL, Invitrogen, Carlsbad, CA, USA) and maintained in a 37 °C, 5% CO_2_ incubator.

#### 4.5.3. Measurement of Viability of RAW264.7 Cells

After dispensing RAW264.7 cells to a final concentration of 2 × 10^5^ cells/mL in a 96-well plate, the plate was incubated for 24 h at 37 °C and 5% CO_2_ and then treated at concentrations of 25, 50, and 100 µg/mL cells, and cultured for 20 h. After injecting 10 µL of Quanti-MaXTM into the plate, the plate was incubated for 4 h. Absorbance was measured at a wavelength of 450 nm using an ELISA reader (Tecan Group Ltd., Mannedorf, Switzerland).
Cell viability (%) = B/A × 100

(A: absorbance of the sample-free group; B: absorbance of the sample-added group).

#### 4.5.4. Nitrite (NO) Measurement

RAW264.7 cells were dispensed into a 96-well plate at a final concentration of 2 × 10^5^ cells/mL. The plate was then incubated at 37 °C with 5% CO_2_ for 24 h. After the incubation period, the extract was processed. Following 1 h of treatment with LPS (1 µg/mL), the cells were further incubated for 16 h. Subsequently, 100 µL of cell culture medium and 100 µL of Griess reagent were mixed and incubated at room temperature for 15 min. The absorbance was measured at 540 nm using a microplate reader. The obtained absorbance values were applied to a standard curve with sodium nitrate to calculate the amount of NO produced.

#### 4.5.5. Western Bolt Analysis

RAW264.7 cells were dispensed into a 6-well plate at a final concentration of 2 × 10^5^ cells/mL and incubated at 37 °C with 5% CO_2_ for 24 h. The samples were treated with a concentration of 50 μg/mL. After 1 h, LPS was added at a concentration of 1 μg/mL and incubated for either 24 h or 30 min. The cells were then centrifuged twice with PBS, and RIPA buffer was added to extract proteins. The protein concentration was quantified by measuring the absorbance at 595 nm using the Bradford protein assay reagent. The quantified proteins were separated by SDS-PAGE and transferred to a polyvinylidene difluoride (PVDF) membrane at 100 V for 1 h. The membranes were blocked with 5% BSA in Tris-buffered saline with 1% Tween-20 (TBST) at room temperature, washed three times for 10 min with TBST, and then incubated with primary antibodies (such as anti-HO-1, anti-iNOS, anti-COX-2, anti-NF-κB, anti-β-actin) at 4 °C for 16 h. After washing with TBST three times for 10 min each, secondary antibodies (goat anti-rabbit IgG HRP, rabbit anti-mouse IgG HRP) was added and incubated in 5% BSA for 2 h at room temperature. The membranes were then washed three times with TBST for 10 min each. Images were obtained using an ultraviolet (UV) imaging system.

#### 4.5.6. ELISA Assay

After dispensing RAW264.7 cells into a 6-well plate at a final concentration of 2 × 10^5^ cells/mL, the plate was incubated at 37 °C with 5% CO_2_ for 24 h. The samples were then treated with a concentration of 100 μg/mL. After 1 h, the cells were further treated with 1 μg/mL of LPS and incubated for 24 h. The supernatant was collected for further analysis. Cytokine levels were measured using an ELISA kit from R&D Systems.

### 4.6. Statistical Analysis

All experimental values are expressed as mean ± standard deviation (mean ± SD), and IBM SPSS Statistics 22 (IBM, New York, NY, USA) was used for statistical analysis. Comparisons between experimental groups were performed by one-way analysis of variance (ANOVA), and Duncan’s multiple range test was performed as a multiple comparison methods to determine significant differences between the experimental groups. Significant differences occurred at a *p*-value of <0.05.

## 5. Conclusions

The study focused on investigating the quality, antioxidant activity, and anti-inflammatory activity of adlay sprouts, with different trends observed depending on the harvesting time. Here are the key findings derived from Table 5: in terms of quality, there was a strong positive correlation (r = 0.99) observed between the harvesting time and the length of adlay sprouts as well as their production mass. However, for economic efficiency and yield increase index per unit area, it is advantageous to harvest adlay sprouts on the fifth to seventh day after sowing; (2) if the optimal harvesting time is considered from the perspective of coixol content, there was a strong negative correlation observed between the coixol content and the production mass (r = −0.97). Therefore, it can be concluded that an early harvesting time would be advantageous for obtaining higher levels of coixol when the objective is to secure bioactive compounds; (3) the antioxidant activity indicators, such as DPPH, ABTS radical scavenging capacity, and total polyphenol contents, exhibited significant correlations with each other, and a higher harvesting time showed a tendency towards increased antioxidant activity. While no clear correlation was observed among the anti-inflammatory activity indicators such as NO, IL-6, and PGE2, it is considered important for producers to select the harvesting time according to their target goals when harvesting adlay sprouts. These findings support the emerging market for sprouts and highlight the potential for adlay sprouts to expand beyond their current seed-centric market. This can open up new opportunities in the food industry and have a significant impact on food safety and toxicity considerations. Further studies are recommended to explore the cytotoxicity and safety of adlay sprout consumption in humans, both in vitro and in vivo, to ensure its suitability as a food ingredient and its potential health benefits.

## Figures and Tables

**Figure 1 plants-12-02975-f001:**
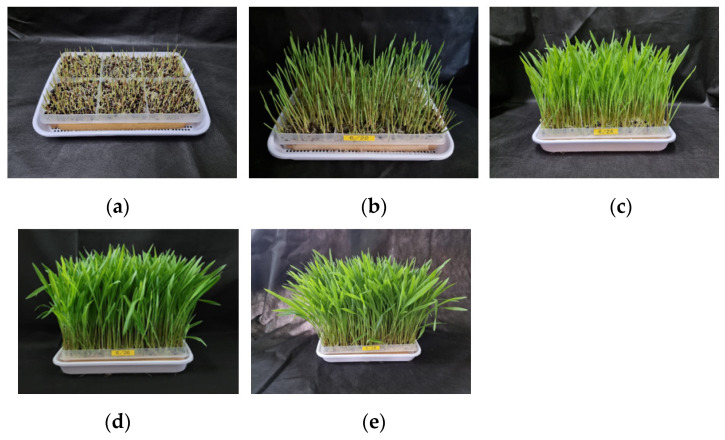
Growth appearance of adlay (*Coix lacryma-jobi* L. var. *ma-yuen* Stapf.) sprout according to the days after sowing. (**a**) Adlay sprout of 3 days after sowing; (**b**) Adlay sprout of 5 days after sowing; (**c**) Adlay sprout of 7 days after sowing; (**d**) Adlay sprout of 9 days after sowing; (**e**) Adlay sprout of 11 days after sowing.

**Figure 2 plants-12-02975-f002:**
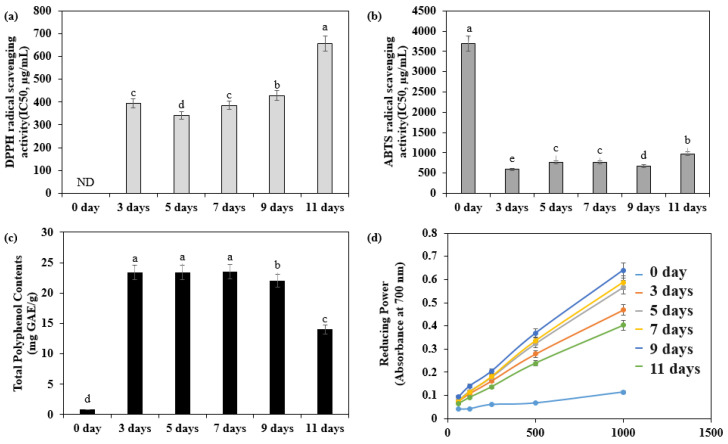
Antioxidant capacity and total polyphenol contents of adlay (*Coix lacryma-jobi* L. var. *ma-yuen* Stapf.) sprout according to the days after sowing. (**a**) 2,2-Diphenyl-1-picryl-hydrazyl-hydrate (DPPH) radical scavenging activities of adlay sprout extract (ASE) according to the days after sowing; (**b**) 2,2′-azino-bis (3-ethylbenz-thiazoline-6-sulfonic acid) (ABTS) radical scavenging activities of adlay sprout extract (ASE) according to the days after sowing; (**c**) Total polyphenol contents of adlay sprout extract (ASE) according to the days after sowing; (**d**) Reducing power of adlay sprout extract (ASE) according to the days after sowing. Values are presented as means ± standard deviation. Same letters are not significantly different by Duncan’s Multiple Range Test (DMRT) and *p* = 0.05.

**Figure 3 plants-12-02975-f003:**
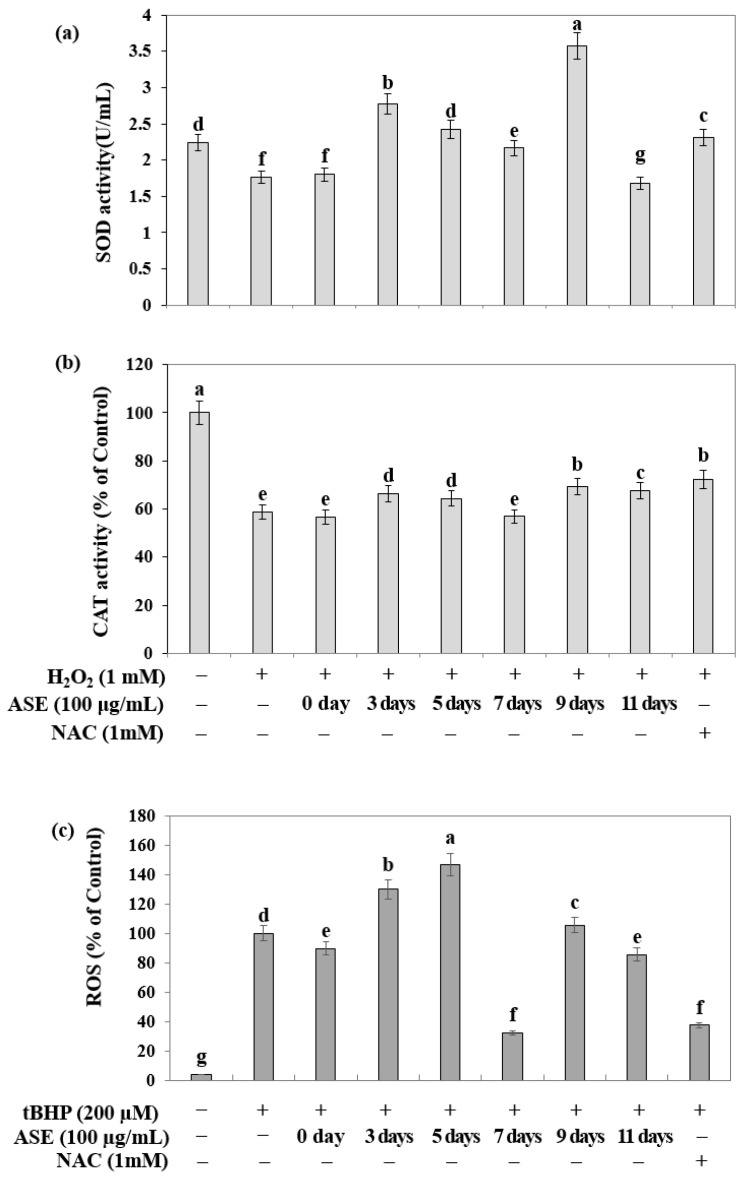
Intracellular antioxidant capacity of adlay (*Coix lacryma-jobi* L. var. *ma-yuen* Stapf.) sprouts according to the days after sowing. (**a**) SOD activity of adlay sprout extracts (ASE) by harvest time; (**b**) Catalase activity of adlay sprout extracts (ASE) by harvest time; (**c**) ROS scavenging activity of adlay sprout extracts (ASE) by harvest time. Values are presented as means ± standard deviation. Same letters are not significantly different by Duncan’s Multiple Range Test (DMRT) and *p* = 0.05. (−); untreated experiment, (+); treated experiment. NAC (*N-acetylcysteine*) was used as a positive control.

**Figure 4 plants-12-02975-f004:**
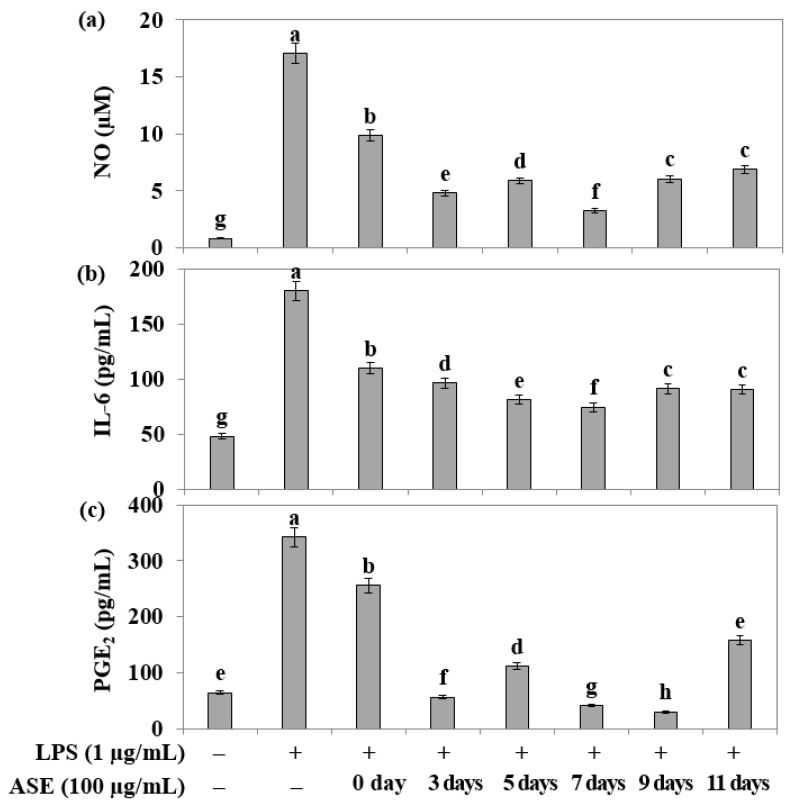
Effects of adlay sprout extract (ASE) on LPS-induced NO, IL-6, and PGE2 production in RAW 264.7 cells. Cells were treated with 100 μg/mL of adlay sprout extract (ASE) 1 h before a 24 h LPS treatment. (**a**) Total NO production was determined using Griess reagent, and a standard curve was created using NaNO_2_ in culture medium. Control values were obtained in the absence of LPS or adlay sprout extract (ASE). (**b**,**c**) Following the manufacturer’s instructions, levels of PGE2 in the media were detected using a specific enzyme immunoassay. Values are means ± standard deviation (*n* = 3), and those with different alphabet are significantly different at *p* < 0.05 by Duncan’s Multiple Range Test (DMRT) treated with lipopolysaccharide (LPS) in the absence of adlay sprout extract (ASE). (−); LPS-untreated experiment, (+); LPS-treated experiment.

**Figure 5 plants-12-02975-f005:**
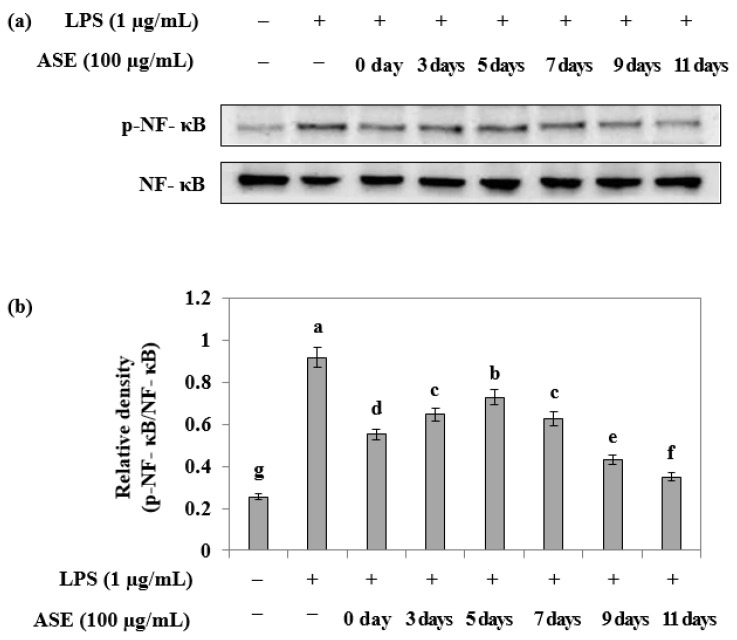
Effect of adlay sprout extract (ASE) on LPS-induced NF-kB activation in RAW264.7 cells. (**a**,**b**) Cells were treated adlay sprout extract (100 μg/mL) and LPS (1 μg/mL) for 16 h. p-NF-kB and NF-kB protein expression were analyzed by Western blotting. Data are expressed as the mean ± SD of three independent experiments and were analyzed by one-way ANOVA. (−); LPS-untreated experiment, (+); LPS-treated experiment.

**Figure 6 plants-12-02975-f006:**
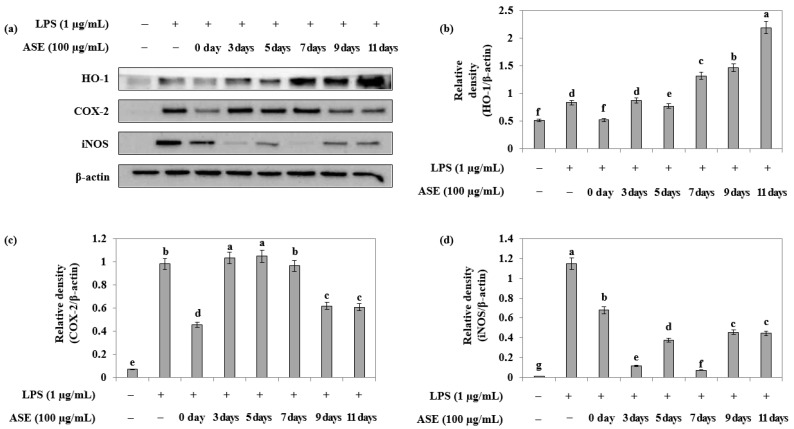
Effect of adlay sprout extract (ASE) on LPS-induced HO-1, COX-2 and iNOS protein expression in RAW264.7 cells. Cells were treated adlay sprout extract (100 μg/mL) and LPS (1 μg/mL) for 16 h. (**a**) Protein levels were detected using b-actin expression as an internal control. (**b**) HO-1 levels; (**c**) COX-2 protein levels; and (**d**) iNOS protein levels were detected using b-actin expression as an internal control. Data were presented as the mean ± SD of three independent experiments and were analyzed by one-way ANOVA. (−); LPS-untreated experiment, (+); LPS-treated experiment.

**Figure 7 plants-12-02975-f007:**
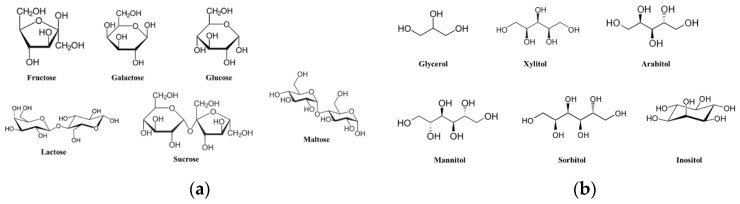
Analysis structure of free sugars and sugar alcohols. (**a**) Free sugars; (**b**) Sugar alcohols.

**Table 1 plants-12-02975-t001:** Growth characteristics of adlay (*Coix lacryma-jobi* L. var. *ma-yuen* Stapf.) sprout according to the harvest time.

Days after Sowing	Sprout Length (cm)	Sprout Weight (g/Plant)	Production Mass (g/cell)	Coixol Content (mg/g)	L (Lightness)	A (Redness)	B (Yellowness)
0 day	-	-	-	0.11 f	-	-	-
3 days	2.75 e ^1^	0.05 e	161.4 e	39.38 a	52.62 a	−0.17 a	8.07 ns
5 days	10.06 d	0.16 d	447.8 d	30.03 b	50.84 ab	−1.72 b	9.12
7 days	15.01 c	0.21 c	814.0 c	27.08 c	48.09 b	−2.17 c	7.71
9 days	18.67 b	0.27 b	1100.9 b	19.41 d	48.43 b	−2.45 cd	7.79
11 days	21.87 a	0.32 a	1319.1 a	19.08 e	47.95 b	−2.81 d	8.22

^1^ Same letters are not significantly different by Duncan’s Multiple Range Test and *p* < 0.05. ns, not significant.

**Table 2 plants-12-02975-t002:** Free sugar contents of adlay (*Coix lacryma-jobi* L. var. *ma-yuen* Stapf.) sprout according to the harvest time.

Days after Sowing	Fructose (mg/g)	Galactose (mg/g)	Glucose (mg/g)	Lactose (mg/g)	Sucrose (mg/g)	Maltose (mg/g)
0 day	35.37	1.14	3.92	1.79	96.27	N.D. ^1^
3 days	17.28	3.68	74.66	0.9	30.68	N.D.
5 days	24.8	3.9	68.49	0.46	13	N.D.
7 days	24.41	3.54	55.33	0.34	8.95	N.D.
9 days	22.1	3.67	51.48	0.48	12.82	N.D.
11 days	18.73	2.3	25	0.34	9.29	N.D.

^1^ N.D., not detected.

**Table 3 plants-12-02975-t003:** Sugar alcohol contents of adlay (*Coix lacryma-jobi* L. var. *ma-yuen* Stapf.) sprout according to the harvest time.

Days after Sowing	Glycerol (mg/g)	Xylitol (mg/g)	Arabitol (mg/g)	Mannitol (mg/g)	Sorbitol (mg/g)	Inositol (mg/g)
0 day	6.49	N.D. ^1^	0.1	0.2	0.22	1.86
3 days	2.48	N.D.	N.D.	0.11	0.04	1.72
5 days	2.16	N.D.	N.D.	0.11	0.04	1.96
7 days	2.08	N.D.	N.D.	0.16	N.D.	2.17
9 days	2.57	N.D.	N.D.	0.18	N.D.	2.89
11 days	2.26	N.D.	N.D.	0.2	N.D.	2.13

^1^ N.D., not detected.

**Table 4 plants-12-02975-t004:** Analytical conditions of HPLC for analysis of adlay sprout extract.

Parameters	Condition		
Column	YMC-Triart C18 (250 mm × 4.6 mml.D., S-5 μm, 12 nm)		
Detection	230 nm		
Flow rate	0.8 mL/min		
Column temp.	40 °C		
Solvent A	0.1% formic acid in water		
Solvent B	0.1% formic acid in Acetonitrile		
Gradient	Time (min)	% A ^1^	% B ^2^
	2	100	0
	45	50	50
	50	5	95
	55	5	95
	55.1	100	0
	60	100	0

^1^ 0.1% formic acid in water; ^2^ 0.2% 0.1% formic acid in acetonitrile.

**Table 5 plants-12-02975-t005:** Correlation coefficient analysis among the indicators of adlay sprout’s quality, antioxidant capacity, and anti-inflammatory activity.

	Sprout Length	Production Mass	Coixol Content	DPPH	ABTS	TPC	NO	IL-6	PGE_2_
Sprout length	1								
Production mass	0.987914	1							
Coixol content	−0.9818	−0.96947	1						
DPPH	0.625228	0.698715	−0.55066	1					
ABTS	0.733109	0.717398	−0.64365	0.776674	1				
TPC	−0.67926	−0.72909	0.617008	−0.97191	−0.87126	1			
NO	0.369828	0.393757	−0.43963	0.524752	0.491807	−0.6495	1		
IL-6	−0.16259	−0.04488	0.090272	0.302567	−0.27653	−0.27527	0.572209	1	
PGE_2_	0.279041	0.263639	−0.23423	0.486651	0.822229	−0.71894	0.633928	0.049668	1

## Data Availability

Data are contained within the article and Appendix A.

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
