# Peer review of "Comparison of Quality, Antioxidant Capacity, and Anti-Inflammatory Activity of Adlay [Coix lacryma-jobi L. var. ma-yuen (Rom. Caill.) Stapf.] Sprout at Several Harvest Time"

_plants, 2023, doi:10.3390/plants12162975_

Round 1
Reviewer 1 Report
Sprout vegetables are considered an excellent health food and commercial consumers are increasing fast. So, the article is welcomed based on the sprout vegetables interest in food industry. Furthermore, the medical effects presented in the article were proved in a very good experimental design and scientific presentation.
My only advice for the authors is that an multivariate analysis should emphasize the classification of sprout quality over time stamp. In this way one can prescribe the best harvest time that provides the best functional food quality and desired medical effects.
Author Response
I sincerely appreciate your careful review and assessment of my paper. Sprouts are increasingly recognized as excellent health foods, as you mentioned, and their commercial consumption is rapidly growing. You requested me to provide an analysis method to ensure the highest functional food quality and efficacy. Therefore, I conducted a correlation analysis of quality indicators based on the harvesting time, and I have interpreted the findings in the conclusion section. The interpretation of the findings in accordance with that is as follows.
In terms of quality, there was a strong positive correlation (r=0.99) observed between the harvesting time and the length of adlay sprouts as well as their production mass. However, for economic efficiency and yield increase index per unit area, it is advantageous to harvest adlay sprouts on the 5th to 7th day after sowing; (2) If the optimal harvesting time is considered from the perspective of coixol content, there was a strong negative correlation observed between the coixol content and the production mass (r=-0.97). Therefore, it can be concluded that an early harvesting time would be advantageous for obtaining higher levels of coixol when the objective is to secure bioactive compounds; (3) The antioxidant activity indicators, such as DPPH, ABTS radical scavenging capacity, and total polyphenol contents, exhibited significant correlations with each other, and a higher harvesting time showed a tendency towards increased antioxidant activity. While no clear correlation was observed among the anti-inflammatory activity indicators such as NO, IL-6, and PGE2, it is considered important for producers to select the harvesting time according to their target goals when harvesting adlay sprouts. These findings support the emerging market for sprouts and highlight the potential for adlay sprouts to expand beyond their current seed-centric market. This can open up new opportunities in the food industry and have a significant impact on food safety and toxicity considerations. Further studies are recommended to explore the cytotoxicity and safety of adlay sprout consumption in humans, both in vitro and in vivo, to ensure its suitability as a food ingredient and its potential health benefits.
Reviewer 2 Report
Comparison of quality, antioxidant, and anti-inflammatory ac- 2
tivity of sprouted adlay [Coix lacryma-jobi L. var. ma-yuen 3 (Rom. Caill.) Stapf.] by harvest time is an interesting paper with a differente approach of evaluating functional properties. However, authors must achieve some modifications to the manuscript in order to improve the quality of the paper.
The abstract does not reflect the content of the paper, does not summarize the methodology used and the purpose of the research is not understood.
The objective in the introduction is not congruent with the content of the paper in general.
Regarding the methodology, Figure 7 is not included in the document.
Regarding the GC/MS analysis, how could the authors know or compare the molecular structures obtained? Were they compared with any library or were internal standards used? Please clarify this part.
In the discussion of results it is important that the authors highlight why the use of multiple methods of antioxidant determination and their possible correlation that all these results have with each other.
Author Response
To improve the quality of the paper, I have prepared a revision manuscript incorporating the suggestions provided by the reviewer. I have made the abstract more readable by making necessary modifications. In the introduction section, I have clarified ambiguous expressions and revised the references. I have also revised the content related to Figure 7. Additionally, I have included information about GC/MS analysis based on the library reference. The discussion section has been reworked to enhance the presentation of the highlights, and overall content has been revised for better clarity
Reviewer 3 Report
Manuscript ID: plants-2351727
Dear Authors,
In this study, sprouted adlay was harvested at several harvesting time, and its productivity, functional components, antioxidant power, and inflammatory activity have been reported.
The current work has several issues and non-acceptable in the current form. Therefore, I would like to suggest for rejection. There are some of my comments and recommendations below in order to solve the current issues and to improve the quality of current work and make it suitable for the possible resubmission.
The title:
- Please add (activity or power) after antioxidant.
- Instead of “by harvest time” could be replaced by “at several harvest time”.
Abstract:
- The abstract should ideally be structured according to the IMRaD format (Introduction, Methods, Results & Discussion, Conclusions). Provide a structured abstract if possible. Current abstract is non-accurate and general. Therefore, rewrite the abstract, my recommendations are listed below:
- L 18: instead of “sprouted adlay was harvested some days after sowing” it is better to write the main approach exactly as follow: sprouted adlay was harvested at several periods i.e., 3, 5, 7, 9, and 11 days after sowing.
- Result part (L 20- L 30) is non-effective, please, add numerical findings and p value of these texts.
- L 23: Add the full word before any abbreviated word first time mentioned, such as COX-2 to be as follow: cyclooxygenase 2 (COX-2). Apply this issue for the similar cases throughout the manuscript.
- L 23, and L 27: Why Anti-inflammatory activity mentioned in various places? it should be merged.
- L 30-32: conclusion part is general, please revise and focus on the main direct findings and state the future perspectives.
Introduction:
- L 44-66: this part is general and should be removed or summarized. The focus should be on sprouted adlay no other plants.
- L 56: you should write [13-15].
- L 57: what you mean “rich in functionality”? L 57- 58 require a revision for better understanding.
- L 64: add two/three references about the adlay origin, chemical composition before the phytochemical substances.
- L 67: “adlay sprouts have been investigated for in vitro anticancer activity” justify and write the main findings, which type of cancer was investigated?
- L 67-69: write in more details and more specifications about bioactive substances of sprouted adlay, these sentences are non-informative and general.
- L 70-78: is very confusing? What is the relation between adlay sprouts and reddish sprouts?? Also, chemical composition, health benefit etc. of adlay seeds/ leaves/extracts etc. should come before the sprouted adlay and their biological and health activities (from general to specific).
· For introduction; you should follow the scientific phenomenon and law by narration the general literatures to be focused on the adlay sprouts and finish by the research gap and the objectives.
- L75: TPC must be defined first time mentioned, apply this issue for all abbreviations all throughout the manuscript.
- L 81: are you sure about this word is correctly placed “radish” ?? in “Germinated adlay radish water extract”.
- L 84-86: the aim should be clarified better and exactly and should be matched with the title of the manuscript! And the specific variety of adlay investigated should be mentioned What is the research gap?
Materials and methods:
- I feel disappointed about the authors didn’t write the references of the methods used in this study to reproduce this work!?
- Overall, the material and methods section are insufficient—lack of accuracy, and requires more details and justification and the model, manufacturers and country should be added to the reagents, enzymes, mediums as well the instruments in order to reproduce this research. Note that, the model should be mentioned first time only don’t repeat it, moreover, take care with the abbreviations.
- L 75: I am confused what is the exactly investigated variety? in title “Coix lacryma-jobi L. var. ma-yuen (Rom. Caill.) Stapf.”? In figure 1 “(Coix lacryma-jobi L. var. ma-yuen Stapf”, while in methods “The seed of the ‘Johyun’ variety of adlay” in line 281??
- L 147: write the version, city, country of “PAST software program”.
- L 281: add the season of the seed production.
- L 284: “then dried for 48 h” Where the drying process carried out? Write the model if you are used an oven.
- L 287: “following recommendations from previous studies” specify and cite these studies?
- L 291” the sprouted adlay extract was diluted in 70% EtOH at a ratio of 1:40” this sentence needs a revision? Are you extracted sprouted adlay by 70% ethanol? What about aqueous extract? Why you used ethanol? Also, how you mentioned the extraction before the dissolving? The extraction should be written in details! Moreover, specify the expression of 1: 40 w/v?
- L 292: write the model of Lyophilizer
- L 296: “Sprouted adlay extract lyophilized powder was re-dissolved in methanol” while you extracted the bioactive compounds of sprouted adlay in 70 % ethanol, however, you resolved in methanol? How? You should write the reference of the extraction and method of Coixol content analyses.
- L 297: ml must be mL, apply for the similar cases.
- L 300: “water (A)-(0.1% formic acid) and acetonitrile (B)-(0.1% formic acid)” is non-clear!
- Table 1 has several errors e.g., write HPLC model, city country, ㎖? (Should be mL), specify the detector used, define the abbreviations under Table note such as ACN, SOL B, remove word table 1 under the table, in the Gradient part where is solvent A concentration versus solvent B??
- L 309: remove 1 Tables may have a footer!
- Write the reference “4.2.2. Chromaticity measurement” and “4.2.3. Analysis of free sugar and sugar alcohol content
- L 331: instead of Analysis structure write chemical formula.
- L 343: The authors didn’t follow author guide for reference style “by Blois (Blois, 1958).” Additionally, this ref. is very old, cite a recent one.
- L 354: correct the name as follow: ABTS (2,2'-azino-bis (3-ethylbenzothiazoline-6-sulfonic acid)).
- Also add space between the section number and the head name (line 354).
- L 357: Very confusing, has errors “absorbance at 732 nm was ± 0.04 (± standard error)”??
- L 360 “was measured at 732 nm” write the details of instrument used (ELISA).
- L 377: what do mean “the sample was dissolved in DW”? dry weight? Or distilled water? DW should not abbreviated! Write the model of water distillator.
- L 377, I am confused, why you resolved in water the sample, while you extract them by ethanol??
- L 381 “centrifuged at 1200 rpm” write the model of centrifuge, also convert rpm to xg, also this speed I think is low!
- L 386: write the full form of ROS.
- L 388: write the type and model of incubator!
- L 38+: What do mean “the samples were treated at a concentration of 100 µg/mL”? which solution?
- L 392: specify the incubation temperature.
- L 409: The source of HepG2 should be at the first time mentioned not here!
- L 410: write the manufacturer and country of the DMEM medium. Also, DMEM abbreviation should be written as full form first time mentioned (no in line 422).
- L 418: You must not repeat the ELISA model several times!
Results and discussion:
- L 100: the scientific name of adlay must be italic.
- In Table 1: check the correct expression units of Sprout weight (g/ea) & Sprout yield (g/cell)! And should be matched as you wrote in the text.
- L 106: p value should be revised “P <0.05”! also “Same letters…” these letters should be mentioned, also add ns mean under table caption.
- Table 2: also, for all tables: specify the full form of abbreviated word under table caption. Table 2 & 3 is “ND”.
- L 127-129: why you inserted toxicity results under antioxidant activity? It should under a separate section.
- L 130: why reducing power is capital?
Conclusion:
-This sentence requires revision “This would help to move beyond the existing seed-centered adlay market and expand into a new food market to significantly impact food safety and toxicity in the future”.
-Also, more studies both in vitro and in vivo about the cytotoxicity and safety of sprouted adlay to human health as food should be carried out. (You should write this sentence as recommendation).
References:
- Double check the references format according to journal guide.
- All strains and scientific names must be italic e.g., Hordeum vulgare (line 513), Cicer arietinum (line 523).
Specific comments:
- The authors are invited to make more efforts and substantial modifications on the current version.
- The language must be improved since the current level is very bad. I recommend a native speaker to revise the manuscript.
- Overall; the quality of presentation is very bad throughout the manuscript.
- There are errors in expression of DPPH and ABTS results (line 133), they should be % not µg/mL as you wrote. Also as stated in methods DPPH, ABTS should be as percentage (see lines 349 and 361). Be note that, only IC 50 to be expressed as µg/mL.
- Some results are non-accurate and requires repeat e.g., antioxidant activities.
- The standard deviation (SD) in the all figures should be revised and generated form the software. Only a part of Fig. 3 is accurate (attached in pdf file).
- For example, in Fig. 2 SD (attached below in pdf file) must not draw manually but through the statistics software automatically.
Good luck!

The manuscript is hard to read. The language must be improved since the current level is very bad. I recommend a native speaker to revise the manuscript.
Author Response
I have addressed the recommended revisions provided by the reviewer in the attached Word document. I sincerely appreciate your careful review and valuable feedback, which have contributed to enhancing the quality of the manuscript. Thank you

Reviewer 4 Report
In my opinion, the article needs no changes and recommends it for publication.
Author Response
I sincerely appreciate your review of my paper. I will make every effort to incorporate the reviewer's feedback and submit an improved version of the paper. Thank you.
Reviewer 5 Report
- to replace in line 469 "ELSIA" with "ELISA".Author Response
I sincerely appreciate your review of my paper. I will make every effort to incorporate the reviewer's feedback and submit an improved version of the paper. I am submitting the paper, incorporating the suggested terminology revisions. Thank you.
Reviewer 6 Report
The manuscript comprises a series de experiments, and by the results and conclusions, all the objetives were considered as satisfactory, for which the manuscript deserves to be published.
Author Response
I am grateful for your review, considering that my manuscript, composed of a series of experiments, has met all the objectives satisfactorily and has value for publication based on the results and conclusions. I will make efforts to submit a high-quality paper.
Round 2
Reviewer 3 Report
Please address the following minor comments:
- Add the quantitative data/numerical in the abstract.
-L 17-21: reduce this content.
- Each abbreviated word must be defined in the figure caption such as SAE in Fig. 3 etc.....
Fine
Author Response
I sincerely appreciate your careful consideration and review of my manuscript. I have added quantitative data to the Abstract section. The numerical data reflects the increasing values from the 3rd day to the 11th day after sowing the sprouts. I have omitted the content of the initial part of the Abstract. Additionally, I have included definitions in the figure captions, such as ASE (adlay sprout extract) in Figure 3, etc. I would be grateful if you could review the attached revision paper. Thank you.
